# Strategies Used in Canadian Nursing Programs to Prepare Students for NCLEX-RN^®^ Licensure Exam

**DOI:** 10.3390/healthcare11040613

**Published:** 2023-02-18

**Authors:** Caroline Gibbons, Isdore Chola Shamputa, Michelle Le, Rose McCloskey

**Affiliations:** 1School of Nursing, Université de Moncton, Moncton, NB E1A 3E9, Canada; 2Department of Nursing & Health Sciences, University of New Brunswick, Saint John, NB E2L 4L5, Canada

**Keywords:** education strategies, NCLEX-RN, nursing education, nursing students, faculty

## Abstract

Nursing educators need strategies for preparing students to be successful in the National Council Licensure Examination (NCLEX-RN^®^). Understanding the educational practices used is an important step in informing curricular decisions and helping regulatory agencies evaluate nursing programs’ efforts to prepare students for practice. This study described strategies used in Canadian nursing programs to prepare students for the NCLEX-RN^®^. A cross-sectional descriptive national survey was completed by the program’s director, chair, dean, or another faculty member involved in the program’s NCLEX-RN^®^ preparatory strategies using the LimeSurvey platform. Most participating programs (n = 24; 85.7%) use one to three strategies to prepare students for the NCLEX-RN^®^. Strategies include the requirement to purchase a commercial product, the administration of computer-based exams, NCLEX-RN^®^ preparation courses or workshops, and time dedicated to NCLEX-RN^®^ preparation in one or more courses. There is variation among Canadian nursing programs in how students are prepared for the NCLEX-RN^®^. Some programs invest considerable effort in preparation activities, while others have limited ones.

## 1. Introduction

The NCLEX-RN^®^ exam is designed to evaluate the knowledge, skills, and critical thinking needed to practice safely as a new registered nurse. The content included on the exam is determined through a robust practice analysis consisting of the identification of nursing activities and abilities required to provide minimally safe and effective client care. These activities and abilities are determined by a purposive sample of Canadian and American subject matter experts and later validated through a survey distributed to newly licensed registered nurses, registered nurse educators, and registered nurse supervisors [1]. Although the NCLEX-RN^®^ is relatively new in Canada, having been first implemented in 2015, it has been used as the licensure exam in the United States since 1988. As a new exam, nurse educators in Canada are interested in understanding how best to prepare students to be successful in the NCLEX-RN^®^. Of particular interest to Canadian educators is the identification of strategies to support students’ preparation for the NCLEX-RN^®^ that are aligned with existing curricula and are effective [2]. While success on the NCLEX-RN^®^ exam is of paramount importance for students who write the exam and are preparing to enter practice, it is equally important for nurse educators and nursing programs [2].

Student performance on the entry-level practice exam is an indicator of a nursing program’s ability to adequately prepare future nurses for practice [3]. The NCLEX-RN^®^ first attempt pass rate for Canadian-trained candidates increased from 69% in 2015 to 86% in 2020. In contrast, the pass rates for internationally trained candidates decreased from 65% in 2016 to 59% in 2020. Also, the first attempt pass rate for those who wrote the NCLEX-RN^®^ in French only increased from 35.2% in 2016 to 61.8% in 2020 [4]. The priority of nursing programs is to prepare generalist nurses who can demonstrate that they possess the competencies and clinical judgment required to become licensed [5]. As faculty have a vested interest in both students and nursing education, they are justifiably concerned about student performance on the NCLEX-RN^®^ [6].

Historically, nursing programs appeared to rely on trial and error in implementing strategies to meet the learning needs of students in preparation for the NCLEX-RN^®^ exam [7]. With few exceptions [8,9,10], the literature about the NCLEX-RN^®^ in Canada has largely focused on the challenges associated with transitioning to a new licensure exam and the lack of consultation with nursing education leaders about the adoption of the exam [2,11,12]. While these discussions do have merit, they provide little guidance to nurse educators on how to help prepare students to be successful on the licensure exam. Although the literature on NCLEX-RN^®^ preparatory strategies is vast, it is American-based, and to our knowledge, no research has been conducted on strategies used in Canadian nursing programs to prepare students to be successful on the licensure exam. The purpose of this study is to describe strategies used in Canadian nursing programs to prepare students for the NCLEX-RN^®^ exam.

### Background

A review of the literature suggests student preparation for the NCLEX-RN^®^ is a concern for nurse educators. A common theme in this body of literature is the importance of being proactive and implementing strategies that have the potential to increase programs’ performance on the NCLEX-RN^®^ pass rate. Some nursing programs have progression policies that use standardized tests to identify students at risk for failing the licensure exam. Many of these proactive activities are introduced early in the program to allow ample time to address identified concerns [13,14,15,16]. A national study of progression policies notes 64% of nursing programs in the United States allow students to repeat a course only once after obtaining a failing grade, 27% allow students two additional attempts, and 8% allow three or more additional attempts. Nearly 50% of programs report that students who fail a course are not guaranteed a seat in the next course offering, and 21% require students to attend remediation sessions prior to being permitted to re-enroll in the course they failed. These progression policies are based on earlier research and the understanding that students who fail courses are at risk of failing the NCLEX-RN^®^ exam, thereby negatively impacting a nursing program’s overall performance [16].

Other strategies used to prepare students for the NCLEX-RN^®^ exam are a series of NCLEX-RN^®^ style tests and examinations throughout the entire nursing program, benchmarks on exit exams as a requirement for graduation, delivery of exams on computers, including computer adaptive testing, and the use of third-party commercial products, which offer practice in the NCLEX-RN^®^ test question format, the use of computerized testing, immediate feedback, and a report that provides details on performance that help identify areas for content reviews [16,17,18,19,20,21,22]. Faculty development is another strategy identified in the literature. To adequately prepare students for practice, many programs require faculty to maintain expertise in a specialty area, be competent in student assessment, have intimate knowledge of the NCLEX-RN^®^ test plan, and have skills in developing clinical judgment in students and the writing of test questions at higher cognitive levels [16,17,23,24].

There are stark differences in how American and Canadian nursing regulatory bodies respond to NCLEX-RN^®^ results. In the United States, state boards of nursing designate minimum NCLEX-RN^®^ pass rates for graduates of nursing programs. These pass rates range from 75% to 90%, with most states requiring an 80% pass rate. Similarly, the Commission for Education in Collegiate Nursing Education and the Accreditation Commission for Education in Nursing Education set an 80% first-time pass rate standard. Programs that do not achieve these pass rates are at risk of losing their accreditation status and can be required to develop comprehensive plans on how they will address a suboptimal success rate [19]. To date, regulatory nursing bodies in Canada have not established passing standards. Although pass rates are publicly available at the provincial level [3] and are evaluated during the accreditation process [25], sanctions have yet to be implemented for programs with pass rates that fall below the national average.

It has been suggested that there are differences in how American and Canadian nursing programs prepare students for the licensure exam [8]. The fact that the NCLEX-RN^®^ is a relatively new licensure exam in Canada may account for this difference. It has been eight years since the implementation of the NCLEX-RN^®^ as the national licensure exam in Canada, and it is time to identify best practices related to the exam. Given the focus to date on the literature on the NCLEX-RN^®^ within the Canadian context, it is difficult to know with any degree of certainty the utility of the existing American literature. Understanding the educational approaches used in Canada to prepare students for the NCLEX-RN^®^ exam is an important first step in knowing how to support students for the exam. Such information can be used to inform curricular decisions aimed at fostering student success and can assist regulatory bodies in evaluating nursing programs’ efforts to improve NCLEX-RN^®^ performance. The research question is: What are the strategies used in Canadian nursing programs to prepare students for the NCLEX-RN^®^ exam?

## 2. Materials and Methods

### 2.1. Study Design

A cross-sectional design was used to gain an understanding of how accredited nursing programs in Canada prepare students for the NCLEX-RN^®^ exam.

### 2.2. Population

Our goal was to survey every nursing program listed on the Canadian Association of Schools of Nursing (CASN) list of accredited nursing programs in Canada (www.casn.ca, accessed on 1 February 2023). The website of every accredited nursing program was searched for the name and email address of the program’s director, chair, or dean. Of the 103 accredited nursing programs, nine were located in Quebec, which does not require the NCLEX-RN^®^ for licensure, and 34 were considered duplicates as they were identified as alternate streams to a listed nursing core program, such as advanced standing programs, bridging programs, or satellite campuses (Figure 1). After screening, the population of interest was determined to be 60 nursing programs. Of the 60 identified programs, we were able to identify the name and contact information for 58 of them, and an email invitation was sent to every dean, program director, and chair. This group was selected because of their positions and responsibilities in an accredited pre-licensure nursing program in Canada. It was assumed that this group would have first-hand knowledge of how faculty prepare students in their programs for the NCLEX-RN^®^ exam. It was possible for invited participants to extend the survey link to another faculty member who was more knowledgeable of the program’s NCLEX-RN^®^ preparatory strategies; however, it was only possible for the survey to be completed once per program. The inclusion criteria were: (i) faculty member of an accredited Canadian nursing education program and (ii) programs located in a province where NCLEX-RN^®^ is required for licensure. Satellite programs of those meeting the inclusion criteria were excluded from the study.

### 2.3. Instrument

The survey was completed using LimeSurvey. This password-protected platform was used to create, distribute, and collect the survey data. The questionnaire developed by the researchers included four questions about program demographics and eleven questions about activities used to prepare students for the licensure examination and how NCLEX-RN^®^ results are used to inform program and student evaluations. Multiple-choice questions pertaining to NCLEX-RN^®^ preparatory activities asked about the use of commercial products, class instruction, and NCLEX-RN^®^ assessments. Multiple response questions about program evaluation focused on summative and formative evaluation and the use of Mountain Measurement Program Reports (Table 1). These reports provide information on the performance of graduates of a nursing program on the NCLEX-RN^®^ and compare these results to those of other nursing graduates provincially, nationally, and internationally. The questionnaire was piloted and assessed for face validity with four program directors, chairs, or deans and refined to improve the clarity of questions prior to distribution. Data obtained from this pilot were not included in the study results. The survey was available in both English and French, and participants could select the language of their choice.

### 2.4. Data Collection

An email invitation was sent to one dean, director, or chair in each of the identified accredited Canadian nursing education programs in February 2021. Reminder emails were sent after 2 and 4 weeks to those who did not respond to the original invitation. The email invitation contained a link to the English and French versions of LimeSurvey. When potential participants clicked on the link to complete the survey, they were brought to an information page and informed that, if they agreed to participate in this study, they should click on the link to complete the survey. Clicking on the survey link was considered consent to participate in the study. At no time was the respondent required to specify any personal or program-specific data, such as their name or geographical location. The data collected was password-protected, and only the four researchers had access to it. There was no compensation provided to study participants.

### 2.5. Data Analysis

Survey data was exported directly into Excel 365. Descriptive statistics were used to describe the study’s data. Categorical data were summarized using frequencies and percentages. Continuous data were summarized using means and standardized deviations. All partial responses were included in the analysis. Open-ended responses were collated and summarized.

## 3. Results

Of the 58 email invitations, 28 completed the survey, for a response rate of 48.3%. Of those who responded, 24 (85.7%) responded in English and 4 (14.3%) in French. The annual number of students graduating from the programs ranged from 48 to 350 (mean = 146.6; SD = 93.7), with 15 identifying as offering a 4-year nursing degree, eight an advanced standing degree, four licensed practical nursing bridging programs, and five identifying as offering "other types of programs" but not specifying the type of program. Six (21.4%) respondents did not answer this question.

Most respondents (n = 24; 85.7%) indicated that their program uses multiple strategies to prepare students, with the number of strategies ranging from one to three. Strategies used to prepare students included requiring students to purchase a third-party product (i.e., PassPoint, HESI, or U-World) (n = 12; 42.9%), administering computer-based tests and exams (n = 10; 35.7%), offering voluntary (n = 8; 28.6%) or mandatory (n = 2; 7.1%) NCLEX-RN^®^ preparatory classes or workshops, and devoting time to NCLEX-RN^®^ preparation in multiple courses (n = 6; 21.4%) or in a designated course (n = 6; 21.4%). Other strategies identified in the open-text option included using an NCLEX-RN^®^ style of questioning in classes, creating NCLEX-RN^®^-style question banks, facilitating study groups and lunchtime preparatory sessions, and providing general information sessions to students on what they can expect from the NCLEX-RN^®^. Four respondents (14.3%) indicated that their program does not offer any specific activities to prepare students for the NCLEX-RN^®^.

Of the programs that indicated the use of commercial products to prepare students, five used HESI, three used U-World, two used PassPoint, and two indicated “other” but did not specify the product. Nearly half of the programs using commercial products required students to purchase these products out of pocket (n = 5), while one program had the cost incorporated into student fees. One program indicated that they used different commercial products in the past but stopped the practice based on student feedback. The remaining six programs did not identify how the cost of these products was covered, and two programs indicated that students are required to successfully pass assessments associated with these products before being eligible to graduate. Five of the programs reported sharing the results of these assessments with faculty, and two programs indicated that faculty are not provided any information on students’ performance in third-party assessments. Five programs indicated that results from third-party products are used in either formative or summative program evaluation, and three programs reported that the results of these assessments are not used in program evaluation.

Twenty-one (75%) respondents purchased the Mountain Measurement program reports, which describe their graduates’ performance on the NCLEX-RN^®^. Only three respondents indicated that the reports are reviewed by the program dean, director, or chair and that the data from the reports is summarized and shared with faculty. Seven respondents indicated that they do not purchase program reports from Mountain Measurement Inc. One respondent indicated that the data described in the Mountain Measurement reports does not provide sufficient information to inform the teaching and learning process, while another stated that they plan to purchase the reports in the future. None of the respondents indicated that faculty have access to the reports.

## 4. Discussion

To our knowledge, this is the first Canadian survey on the activities used to prepare pre-licensure nursing students for the NCLEX-RN^®^. Study results suggest that nursing programs across Canada are engaging in a variety of activities to prepare students for the licensure exam. This finding supports previous reports that many nursing programs adapted their approaches to teaching and program evaluation in response to the introduction of the NCLEX-RN^®^ in Canada in 2015 [12]. While beyond the scope of this study, findings suggest programs’ efforts to prepare students for the NCLEX^®^ are effective as national pass rates on the licensure exam rose from 69% in 2015 to 86% in 2020 [26]. The fact that some programs report not using any activities or supports to prepare students for the licensure exam while others report using multiple supports is notable. Future studies should examine the effectiveness of the activities and supports used to prepare students for the licensure exam.

Evaluating the outcomes of preparatory activities is important given the impact these activities have on faculty and students. For nursing faculty, developing and delivering NCLEX-RN^®^ preparatory activities demands considerable effort, often without any additional resources [12]. For students, the requirement to purchase commercial products, either out-of-pocket or through student fees, can create financial challenges. Mandatory workshops outside of regular instructional time and requirements to pass pre-licensure assessments can place undue stress on students. The impact of placing additional expectations on students should be examined given the attention given to their stress level and psychological well-being in recent years [27,28]. Although it is acknowledged that some stress can be a powerful motivator [29], it may also adversely affect students’ learning [30]. Ensuring the expectations placed on faculty and students with respect to NCLEX-RN^®^ preparation achieve their desired goal and produce an acceptable return on investment is important. Results from this study provide evidence that some programs are evaluating NCLEX-RN^®^ preparatory activities. One program indicated that they used commercial products in the past and, based on students’ feedback, changed vendors and eventually removed the requirement for students to purchase third-party products altogether. There appears to be a need to incorporate outcomes associated with various NCLEX-RN^®^ preparatory activities into programs’ evaluation plans. While third-party products may be effective in preparing students for the licensure exam, it is possible that decisions about product selection are best made with consideration for learning styles and study preferences. If accurate, decisions about which product to purchase may be best made by individual students. Further work in this area is required.

Findings in this study show faculty have limited access to reports outlining student performance on the NCLEX-RN^®^. While some programs indicated they do not purchase the program reports available through Mountain Measurements Inc., those that do appear to place restrictions on how the information contained in them is shared with faculty. None of the respondents indicated that faculty could readily access reports that detail student performance. Yet reports that detail how students perform on the licensure exam can inform many of the day-to-day decisions faculty make around instruction and formative evaluation. While there are questions as to whether first-time pass rates are valid measures of a program’s performance [31], it is undeniable that performance on the licensure exam is important to nurse educators and students. Further, being able to readily access outcome data on past students’ performance on content areas such as human functioning, health alterations, and the health-illness continuum, among others, can provide empirical data and a framework to guide instructional decisions. The risks of providing faculty with direct access to licensure reports should be weighed against the potential benefits. Furthermore, faculty should be encouraged to engage in professional development that supports them in developing expertise in interpreting and transferring knowledge from these program reports to curriculum activities.

### Limitations and Recommendations

Despite the sample size of 28 nursing programs, a response rate of nearly 50% is above the average survey response rate of 20–30%, as reported by Qualtrics [32]. Results of this study should be interpreted with caution, given the small sample size. The survey was administered during the COVID-19 pandemic, when administrators were dealing with dynamic public health guidelines, faculty needs for support in using alternate modes of delivery for nursing education, and the uncertainties associated with students’ clinical practicums. The response rate observed in this study could also have been a consequence of shifts in technology that occurred even before the pandemic, including increased reliance on surveys for data collection (response burden), security concerns about opening emails from unknown sources, and distrust and stigma associated with requests for program information [33].

Because the survey was anonymous, no program-specific data were available, making it impossible to assess the sample’s representativeness for the population of interest. For example, with the exception of the primary language used to teach in the program, no information was sought on the individual characteristics of programs, such as their size, teaching facilities, or geographical locations. Further, although partial response rates were low, they were included in the data analysis, which must be considered when interpreting the findings.

Based on responses from the invitees who completed the survey, it appeared that preparing students for the licensure exam and helping them be successful is important to nurse educators.

We acknowledge that our email invitation was sent to one dean, director, or chair and that our survey questions were more general and did not focus on strategies that specifically included the clinical practice environment as part of exam preparation. Clinical instruction plays a fundamental role in bringing theory into practice and vice versa. A focus on classroom activities will not be sufficient to prepare students for the next generation NCLEX-RN^®^. Future studies should consider adding specific questions to the survey regarding how students learn and apply decision-making and questions specifically about improving clinical judgment and decision-making skills. Future surveys on preparation activities should consider asking about the role of faculty members, including clinical instructors in the classroom, laboratory, simulation, and clinical experience, in preparing students for the next generation NCLEX-RN^®^, which will take effect in 2023 [34]. Next generation NCLEX-RN^®^ questions are designed to better measure clinical judgment than the current NCLEX-RN^®^ [34]. Nurses need to make more complex decisions, and the next generation NCLEX-RN^®^ follows a model for measuring clinical judgment, which is a combination of critical thinking and decision-making [35]. Clinical instruction also plays a key role in the preparation of students to meet entry-level competencies and test plan learning objectives. It is therefore important for faculty members, including clinical instructors, to be familiar with the National Council of State Boards of Nursing Clinical Judgment Measurement model and the new next generation NCLEX-RN^®^ questions format (e.g., case study, bowtie, trend, etc.) to guide them in incorporating clinical judgment in nursing students as preparation for writing the next generation NCLEX-RN^®^ questions [34]. Faculty, hence, need to help prepare students for an additional type of testing [36]. There is no need to restructure the curriculum to improve the clinical judgment of nursing students, but there is a need to change the focus of the already-established curricular content to incorporate system-level thinking related to the clinical judgment framework [37]. Faculty must shift teaching methodologies to focus on clinical judgment and incorporate it into the classroom, laboratory, simulation, and clinical experiences to nurture a reliance on clinical judgment to prepare nursing students to be successful on the entry-level practice exam and for their transition to the entry-level nurse role [38].

## 5. Implication for an International Audience

The critical role nurses play in the interdisciplinary team and healthcare delivery and the global shortage of registered nurses [39] make passing the licensure exam particularly important. While students can typically write the exam several times, doing so can delay their entry into practice or their ability to assume the roles and responsibilities of a registered nurse in an interdisciplinary team. Although the NCLEX-RN^®^ is the licensure exam in Canada, Australia, and the United States [26], preparing students for practice is relevant for educators worldwide [39]. Internationally, nursing educators need guidance and strategies to prepare students for entry into practice [40], which includes meeting the requirements for registration. Understanding the educational practices used to prepare students for licensure can assist nursing programs in their efforts to increase the supply of nurses amidst the global shortage.

## 6. Conclusions

Findings from this study show variations among Canadian nursing programs in terms of how students are being prepared for the NCLEX-RN^®^. It appears some programs are investing considerable time into the NCLEX-RN^®^ while other programs are limited in their preparatory activities. One area that appears to be particularly lacking is the evaluation of preparatory efforts, including the impact of not using any activities to prepare students for the exam. Given the narrow focus of research conducted on the NCLEX-RN^®^ within the Canadian context, future studies should target the effectiveness of teaching strategies for licensure and how best to use NCLEX-RN^®^ data to improve program performance.

## Figures and Tables

**Figure 1 healthcare-11-00613-f001:**
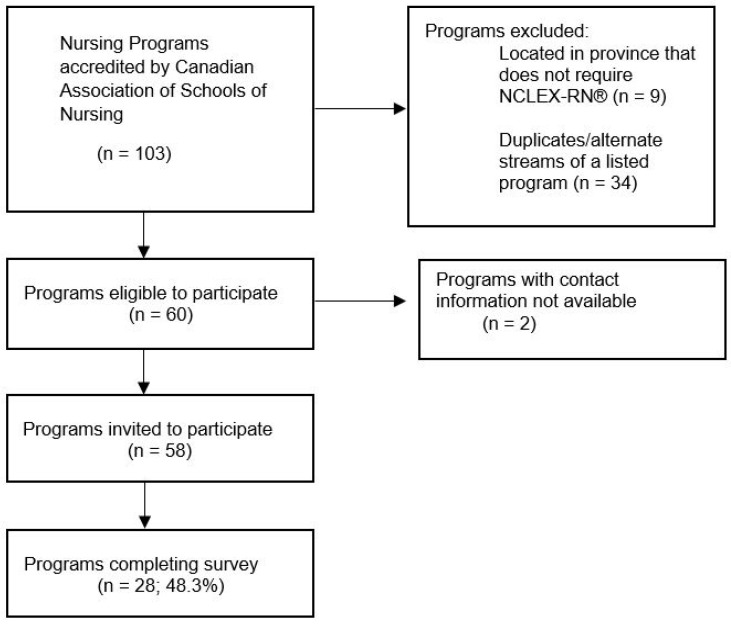
Flow chart of Canadian Nursing Program’s eligibility and participation.

**Table 1 healthcare-11-00613-t001:** Summarized specific survey questions.

Question Number	Questions
1	How does your program prepare students for the license exam?
2	If you require students to use a third-party NCLEX-RN^®^ product, which one do you require?
3	If a third-party NCLEX-RN^®^ program/assessment is used, how is the cost covered?
4	If you require students to use a third-party NCLEX-RN^®^ product, are students required to pass any assessment before graduating?
5	If students are given the opportunity to complete aNCLEX-RN^®^ readiness assessment prior to graduation, how are the results communicated to them? To Faculty?
6	How are the results of the assessment used for program evaluation?
7	How does your program use the Mountain Measurement program reports?
8	Please add any additional comments relevant to student preparation for the NCLEX-RN^®^ exam at your institution.

## Data Availability

The data presented in this study are available on request from the corresponding author.

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
