# Peer review of "Strategies Used in Canadian Nursing Programs to Prepare Students for NCLEX-RN^®^ Licensure Exam"

_healthcare, 2023, doi:10.3390/healthcare11040613_

Round 1

Reviewer 1 Report

Manuscript ID healthcare-2201445: Strategies used in Canadian Nursing Programs to Prepare Students for NCLEX-RN® Licensure Exam Special Issue: Current Nursing Practice and Education

The authors present a cross-sectional descriptive national survey aimed at strategies used in Canadian nursing programs to prepare students for the National Council Licensure Exam (NCLEX-RN®). I agree with the authors that this is an important theme for Canadian nursing, but it’s not clear what sort of gaps or deficiencies in existing studies this particular article is trying to solve. This article might be more suitable in a nursing education journal. In addition, if the editors of Healthcare MDPI decide to move forward with this piece, my strong preference is that the authors provide an explanation of why ethics approval and consent were not sought from participants. 

The authors provide scientific background, an objective statement, and an explanation of the rationale. The introduction is well-developed, however, extensive and long. I suggest shortening it. Strong methodology, however, the overall description of the methods section lacks details. See below a few suggestions/clarifications:

  • Describe the study population in detail (level of education, faculty status, courses taught, etc).   

  • Describe how representative the sample is of the study population (or target population if possible).

  • Was any compensation provided?  

  • Explain why this has not been submitted to an IRB. 

  • Given that the study was not approved by an institutional IRB, please provide information about survey anonymity and confidentiality and describe what mechanisms were used to protect unauthorized access.

  • State how non-response error and missing data were addressed 

Overall the study fails to acknowledge nuances in the Canadian nursing workforce and student profile that can have implications on the NCLEX examination, such as internationally educated nurses and Indigenous students who are often time overrepresented in nursing courses. These are two important intersectionalities that could be explored through equity, diversity, and inclusion lens.

Reviewer 2 Report

We are facing a very interesting study that addresses the different strategies used by educational centers to prepare nursing students to face the NCLEX-RN with guarantees. The approach seems original and appropriate to me, acknowledging the effort of the researchers to capture a sufficiently representative sample.

I recommend the publication of the article, but I want to make some small contributions.

In line #9, the acronym NCLEX-RN is used for the first time without describing it.

It would be interesting to include in the abstract to clarify to whom the study has been addressed, there is no mention of it.

In Material and methods, I consider it pertinent to incorporate a flowchart that graphically describes the scope, the loss of programs and their reasons, until reaching the final 28.

In results, I have a question. We inquired about the success and performance rates of the different training programs? I consider that it would be very interesting data to contrast the effectiveness of the different educational strategies.

Finally, I consider it necessary to incorporate a section on limitations that includes the deficiencies of the research and areas for improvement.
